# Two Probiotic Candidates of the Genus *Psychrobacter* Modulate the Immune Response and Disease Resistance after Experimental Infection in Turbot (*Scophthalmus maximus*, Linnaeus 1758)

Sven Wuertz [1,*], Filipa Beça [1,2], Eva Kreuz [1], Konrad M. Wanka [1,3], Rita Azeredo [4], Marina Machado [4] and Benjamin Costas [4,5]

1 Department Fish Biology, Fisheries and Aquaculture, Leibniz Institute of Freshwater Ecology and Inland Fisheries, Müggelseedamm 310, 12587 Berlin, Germany
2 Faculty of Sciences and Technology, Universidade do Algarve, Campus de Gambelas, Edifício 8, 8005-139 Faro, Portugal
3 Albrecht Daniel Thaer-Institute of Agricultural and Horticultural Sciences, Humboldt University Berlin, 10099 Berlin, Germany
4 Centro Interdisciplinar de Investigação Marinha e Ambiental (CIIMAR), Terminal de Cruzeiros de Leixões, Av. General Norton de Matos s/n, 4450-208 Matosinhos, Portugal
5 Instituto de Ciências Biomédicas Abel Salazar (ICBAS-UP), Universidade do Porto, Rua de Jorge Viterbo Ferreira No. 228, 4050-313 Porto, Portugal
* Correspondence: wuertz@igb-berlin.de

**Abstract:** Probiotic bacteria are a recognized alternative to classical methods of disease prophylaxis and therapy. We tested the effects of their application on the immune reaction in juvenile turbot. To prevent digestion of the probiotics, rectal administration was applied to maximise colonization, by-passing digestion in the stomach. The application of *Psychrobacter nivimaris* and *Psychrobacter faecalis* showed beneficial effects on the inflammatory response and disease resistance after infection with the common pathogen *Tenacibaculum maritimum.* Treatment with *P. nivimaris* and *P. faecalis* resulted in 0% and 8% mortalities post-infection, while in the treatment control, an elevated mortality of 20% was observed. In the challenge controls (no infection), no mortalities were observed during the entire experimental period. After an experimental infection, mRNA expression of selected immune markers (*mhc II α*, *il-1β*, *tcr*, *tgf β* and *tnf α*) were determined by RT-QPCR at 0, 1 and 5 days post-infection (dpi). At 0 dpi, gene expression was comparable between the treatments and the treatment control, suggesting that probiotics did not act via immune stimulation of the host. At 1 dpi, all genes were up-regulated in the treatment control but not in the probiotic groups, indicating that the infection in probiotic-treated fish developed at a less severe level. At 5 dpi, mRNA expression returned to baseline levels. As a conclusion, the native probiotic candidates *P. nivimaris* and *P. faecalis* improved survival, whereas, in the control, mortality increased and expression of the immune markers was up-regulated post infection. This highlights a potential application of *P. nivimaris* and *P. faecalis* in disease prophylaxis, but further research is needed.

**Keywords:** native probiotics; *Psychrobacter*; pathogen challenge; immune markers; survival; gene expression

## 1. Introduction

Turbot is a valuable flatfish species and is particularly interesting for recirculating aquaculture production [1,2]. Indeed, turbot is a cold water adapted species revealing an excellent feed conversion (FCR 0.76–1.2) [3–6], high stress tolerance [7,8], tolerance towards suboptimal water quality [2,9–11], good acceptance of alternative protein sources [5,12–14] and relatively low susceptibility to diseases [15,16]. As such, turbot is a robust fish species with excellent potential for land-based aquaculture.

Aquatic environments represent a constant risk of pathogen transmission. Indeed, bacterial and viral infections are one of the major concerns in aquaculture, destroying entire harvests [17–19]. In the Asian Pacific region, for example, losses of up to 30% of aquaculture yields have been reported [20]. Similarly, in turbot farming, bacterial infections also pose the most serious threat [21] with furunculosis, vibriosis, streptococcosis and tenacibaculosis being the most relevant bacterial diseases [22–26]. Flow-through systems are particularly vulnerable due to the transmission of bacteria from the surrounding environment [27]. If health management including a disinfection unit is effectively set up, a land-based recirculating aquaculture system (RAS) may not be affected to comparable levels. Nevertheless, due to high stocking densities in RAS, transmission from fish to fish may be high and outbreaks are frequently observed, particularly in early life stages [16]. Here, prophylaxis and prevention are the most effective strategies to limit outbreaks, particularly since early life stages cannot be vaccinated [28].

Tenacibaculosis is a bacterial disease induced by *Tenacibaculum maritimum* [29]. It was first described in red and black seabream farming and affects several finfish species worldwide. It is particularly important in turbot farming in Spain and Portugal and numerous outbreaks have been reported [30]. *T. maritimum* is considered an opportunistic pathogen where the outbreak of disease is frequently associated with increasing temperatures, salinity and reduced water quality [30]. The clinical signs in fishes include eroded tissue in mouth, fin and tail, prominent ulcers, and, ultimately, mortality [30,31]. The gills often appear pale, with an increased production of mucus and conspicuous necrotic patches [32]. It has been reported that *T. maritimum* is detected internally, suggesting that it becomes systemic [30]. Flatfish appear to be relatively susceptible compared to other aquaculture species [33,34]. *T. maritimum* shows optimal growth at 30 °C [30], which turns the bacterium into a threat in warm water, for example, in Spain and France. Pathogenesis, transmission, and virulence are still poorly studied [35]. It has been suspected that jellyfish are a natural host and may consequently act as vectors to fish in sea cages [36]. Furthermore, suggested transmission pathways include a host-to-host transmission via seawater and uptake by ingestion [32].

Numerous chemicals, including antibiotics, have been used to assure the health of the farmed fish, but they are problematic for humans and the environment [37–39]. While antibiotic treatment was the classical therapy form in aquaculture, its prophylactic application is forbidden globally. Furthermore, due to the risk of spreading antibiotic resistance, use of antibiotics by the agriculture industry is viewed critically [40,41]. The limitation of antibiotics, increasing antibiotic resistance and the growing consumer concerns require alternative prophylactic and therapeutic strategies [37]. The main strategies of disease control focus on hygiene management and prophylactic treatment, primarily an early vaccination and the application of feed-supplemented immunostimulants such as pro- and pre-biotics. Here, functional diets, which improve gut health, are promising [42]. Most probiotics used in aquaculture originate from terrestrial sources and native probiotics are rarely used on a commercial scale [42]. It has been suggested that these non-native species are less adapted to the host's gut than native (autochthonous) probiotics and, therefore, do not colonize the gut efficiently [42,43].

Most studies on novel probiotics in aquaculture focus on the ability to inhibit growth of selected pathogenic bacteria. This is usually investigated in in vitro assays (e.g., well diffusion agar assays), which identify inhibitory effects on pathogen growth as the utmost important selection criteria [44,45]. However, confirmation of such effects in an experimental challenge is essential. Here, improved resistance towards the pathogen is expressed in higher survival rates of the infected animals [46–50]. Probiotics compete with the pathogens for binding sites for settlement and nutrients, referred to as competitive exclusion. Thereby, probiotics outcompete pathogens and prevent potential infections. In addition, probiotics may stimulate the host's immune system [51,52]. Direct interactions of probiotics with a respective host can increase the expression of immune-related genes and enhance immune responses. This, in turn, can lead to higher survival rates when challenged by pathogens. Similar to conventional immune stimulants such as lipopolysaccharides or ß-glucans, probi-

otic bacteria can induce the expression of MHC-II in antigen-presenting cells [53], triggering an adaptive immune response by binding an antigen and presenting it on the cell surface for T-cell recognition. Similarly, T-cell receptor α (*tcr*) is a key factor in T-cell activation. Probiotic-treated fish also revealed an increased expression of innate-related immune genes such as interleukin 1ß (*il-1β*), tumor growth factor ß (*tgf β*) and tumor necrosis factor α (*tnf α*) [54–56].

Recently, candidate probiotics have been isolated from flatfish including turbot [6,43]. Some isolates showed a promising antagonistic effect against *T. maritimum* in a well diffusion agar assay, including several strains of *Psychrobacter* [43]. These probiotic candidates were evaluated in a classic feeding trial, showing no adverse effects on growth performance or feed conversion in a probiotic mixture in turbot and can, thus, be considered safe [6]. Moreover, no pathogens have previously been described in the genus *Psychrobacter*. *Psychrobacter* are gram-negative rod-shaped bacteria of the gamma-proteobacteria [57]. They are a cold-adapted, osmotolerant, mostly marine species [57]. Several species including *P. faecalis*, have been reported to colonize the gut of finfish species [58]. To date, the genus *Psychrobacter* has been poorly studied. Nevertheless, based on the abundance among candidate probiotics from grouper and flatfish, great potential as a probiotic has been suggested [43,59]. The effects described comprise improved feed utilization [52], positive modulation of the intestinal microbiome [60], modulated immune functions of the host [45,60,61] and antagonistic effects on *T. maritimum* [43].

Therefore, in the present study, two autochthonous candidates *P. nivimaris* and *P. faecalis* previously isolated from wild turbot were used to study the disease resistance towards *T. maritimum* and immune response upon administration. To standardize the administration, avoid deterioration during the passage of the gastrointestinal tract (GIT) and support the colonization in the GIT, we administrated the probiotics via rectal cannulation as previously described [62]. After two applications, we performed an experimental infection with *T. maritimum*, assessing the survival and the mRNA expression of selected immune marker genes.

## 2. Materials and Methods

### 2.1. Isolation and Identification of Probiotic Candidates

Probiotics were isolated from healthy wild turbot and characterised as recently described [43]. Briefly, the two probiotic candidates were identified by 16S rRNA sequencing [6,43]. Subsequently, the GYRB gene was identified with *Psychrobacter*-specific, conserved primers [gyrb F: gAAgT CATgA CCgTT CTgCA CAYgC NggNg gNAAR TTYga; R: AgCAg ggTAC ggATg TgCgA gCCCC RTCNA CRTCN gCRTC NgTCA T] according to Zeng et al. [63,64] supporting an improved identification of the species. This PCR was performed in a 25 μL reaction volume [0.5 U Platinum hot start Taq polymerase (Invitrogen), 2 μL diluted cDNA (20 ng μL$^{-1}$), 1× PCR buffer, 0.4 μM each primer, 3 mM MgCl$_2$, 0.2 mM dNTPs (Qiagen)]. PCR cycling program involved a denaturation at 96 °C for 3 min and 40 cycles of amplification (denaturation 96 °C, 1 min; primer annealing 58 °C, 1 min; elongation 72 °C, 1 min). After direct sequencing, an 1145 bp amplicon of the GYRB gene and identification by sequence comparison, two isolates were selected for the study, *Psychrobacter nivimaris* (Pn) and *Psychrobacter faecalis* (Pf), which inhibited growth of *T. maritimum* in a well diffusion assay [43]. These probiotics were archived in cryoculture according to Wanka et al. [43].

For the experiments, cultures were freshly prepared and stored on ice until used: First, probiotics were plated on BD DifcoTM marine agar 2216. After incubation at 18 °C, a single colony was picked and inoculated in marine broth. After an incubation of 48 h at 18 °C, bacteria were collected by centrifugation (4500× *g*, 30 min, 4 °C), washed and resuspended with 0.9% NaCl (*w/v*). For the experiment, probiotic solutions of 10$^8$ CFU mL$^{-1}$ were prepared by plate counting.

*2.2. Probiotic Treatment*

Healthy, one-year-old juvenile turbot with a mean weight of $30.93 \pm 0.89$ g and a total length of $12.24 \pm 0.18$ cm were acquired from Acuinova S.A. (Mira, Portugal) and transported to the CIIMAR fish facility. One hundred and sixty-two fish were randomly stocked to eighteen rectangular 8 L-tanks (9 fish per tank), which were set up as three separate recirculating aquaculture systems (RAS). Each RAS was composed of tanks and a sump and was run with a water turnover of 2 volumes/h and 2–4% water exchange. The sump included a biological and mechanical filter unit. The water from the tanks was first directed to a mechanical filter, which comprised a series of trays with cloths retaining the debris. Then, the water was directed into a reservoir with the biological filter comprising biocarriers in constant movement and strong aeration. The filter water was then finally pumped into the tanks. The treatment control (C) and two probiotic treatment groups (Pn and Pf) were kept in a separate RAS each (Figure 1). In addition, 30 fish were stocked to another RAS (5 fish per tank) to determine the LD50 of the *T. maritimum* strain. The photoperiod was set according to the natural cycle. The temperature was controlled with a thermostat at $19 \pm 1$ °C. Twice a day, fish were fed a commercial diet (R-3 EUROPA 22%, Skretting) at 1% of the total biomass. Excessive feed and faeces were removed daily to ensure good water quality. Daily, oxygen ($O_2 > 95\%$) and salinity (approximately 35 ppm) were measured with a HQ40d hand probe. Nitrite ($NO_2^-$—N $< 0.02$ mg/L) and total ammonia-nitrogen (TAN $< 0.2$ mg/L) were determined spectrophotometrically with a Palintest 7000SE photometer, using the respective kits. The fish were acclimatized for 30 days.

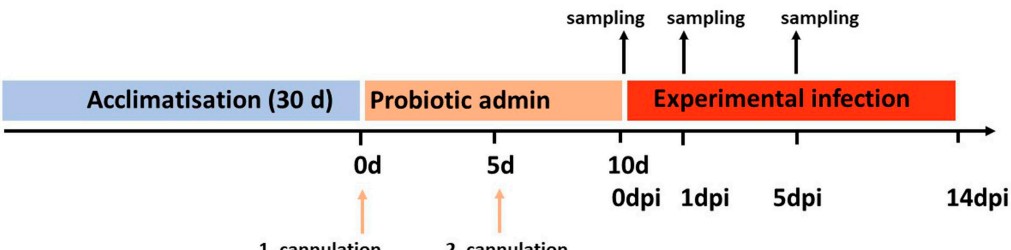

**Figure 1.** Time course of the study comprising an acclimatization (30d), the probiotic treatment with two successive rectal cannulations and the experimental infection with *Tenacibaculum maritimum* ACC6.1. Sampling was performed at 0 days post infection (dpi), 1 dpi and 5 dpi. Mortalities were monitored over 14 dpi.

On day 0 and day 5, fish received two consecutive rectal administrations of the respective probiotic (Pn and Pf) or a 0.9% NaCl solution as treatment control (C). Prior to the rectal administration, fish were anaesthetized by immersing them in an anesthetic bath with 250 ppm 2-phenoxyethanol. The cannula (19 mm length, 0.7 mm diameter) was inserted through the anal pore and 1 mL of probiotic suspension ($10^8$ CFU mL$^{-1}$) or saline solution was slowly injected into the hindgut. Upon removal of the cannula, the anal pore was closed with a finger for 15 s to avoid loss of the suspension. The inoculum was spread throughout the gastrointestinal tract by gentle massage of the body cavity. Feeding was suspended 15 h before the cannulation and resumed approximately 15 h after the treatment. After the treatment, fish were placed in an incubation tank for a couple of minutes to wash off excess inoculum and returned to the respective rearing system. After two applications, we performed an experimental infection with *T. maritimum*, assessing the survival and the mRNA expression of selected immune marker genes. First sampling (0 dpi) was carried out five days after the second rectal cannulation. The experimental infection was carried out directly after this sampling (Figure 1).

*2.3. Determination of LD50 for Experimental Infection*

After acclimation, to determine the LD50, thirty fish (6 tanks, n = 5) were infected with *T. maritimum* ACC6.1 by bath immersion containing $10^6$, $10^7$, $10^8$ CFU mL$^{-1}$ accord-

ing to Avendano-Herrera [65]. The *T. maritimum* strain ACC13.1 was isolated from skin lessions of Senegalese sole in a Portuguese fish farm and was kindly provided by Professor Alicia E. Toranzo (Departamento de Microbiologıa y Parasitologıa, Facultad de Biologıa, University of Santiago de Compostela, Spain). Bacteria were freshly grown on marine agar 2216 [55.1 g L$^{-1}$, pH = 7.6 $\pm$ 0.2, Laboratorios Conda, Spain] at 25 °C for 48 h. A single colony was picked and transferred to marine broth [40.1 g L$^{-1}$, pH = 7.6 $\pm$ 0.2, Laboratorios Conda, Spain] and incubated at 25 °C for 18 h. To prepare the inoculum, exponentially growing bacteria were collected with a Heraeus Multifuge 1 S-R (4500$\times$ *g*, 30 min, 4 °C). The pellet was washed twice with 0.9% NaCl. After resuspension in saline solution, a final concentration of 10$^5$, 10$^6$, 10$^7$ colony forming units (CFU) mL$^{-1}$ was established using a growth curve.

For the infection, the recirculation system was stopped and the water volume of each tank was reduced to a final volume of 2 L in an air-conditioned room (19 $\pm$ 1 °C). Then, the fish were inoculated with the bacteria for 18 h under moderate aeration preventing the formation of foam. Afterwards, the rearing water was changed three times and the recirculation system was restarted.

*Taenacibaculum maritimum* was re-isolated from all dead and moribund fish. Moribund fish with excessive ulcers were sacrificed [66]. The LD50 was calculated according to the method of Reed and Muench [67], identifying the concentration of bacteria applied in the tank that killed 50% of the fish. All fish surviving at the end of the experiments were killed by an anaesthetic overdose (1 mL L$^{-1}$, 2-phenoxyethanol) and examined for *T. maritimum*.

### 2.4. Experimental Infection and Sampling

At 0 dpi, after the first sampling, fish were experimentally infected with freshly prepared *T. maritimum* ACC6.1 at a concentration of 2.65 $\times$ 10$^7$ CFU mL$^{-1}$ as previously described. In addition, challenge controls of non-infected fish (Pfc, Pnc, Cc), that received the cannulation, were reared in a separate RAS. With the exception of this challenge, controls (no infection), which were assessed in duplicate tanks (each n = 5), all the experimental groups were assessed in triplicate. Cumulative mortality was recorded twice a day for a period of 14 days after the experimental infection.

For RT-QPCR analysis, 20–100 mg of spleen tissue were preserved in 500 µL RNAlater (Qiagen), allowed to infiltrate the tissue at 4 °C for 24 h and stored at −20 °C. The study was carried out in compliance with the EU Directive 2010/63/EU and Portuguese legislation and approved by the CIIMAR Animal Welfare Committee (0421/000/000/2020).

### 2.5. Gene Expression

For each data point, total RNA was isolated from 8 individual samples using Trizol [9]. A DNase digestion (Invitrogen, DNAse I amplification grade) was performed to eliminate DNA contamination. RNA later was discharged and samples were washed with 500 µL Trizol to remove any crystals that may have formed during storage. Then, 700 µL of Trizol reagent was added. The tissue was subsequently homogenized with a Qiagen Tissue Lyzer, 600 µL of Trizol were added, samples were vigorously mixed, incubated at RT for 5 min and centrifuged (12,000$\times$ *g*, 10 min, 4 °C). Then, 500 µL of the supernatant were added to 750 µL Trizol in a new vial, mixed and 250 µL of chloroform was added. After mixing, to allow separation of the aqueous phase, samples were incubated for 10 min at room temperature. After centrifugation (12,000$\times$ *g*, 15 min, 4 °C), 300 µL of the upper phase were transferred to a new vial and 300 µL of isopropanol were added to precipitate RNA by incubation for one hour at −20 °C and 3 min at room temperature. RNA was collected by centrifugation (12,000$\times$ *g*, 15 min, 4 °C) and washed with 300 µL of ice cold 70% ethanol. After centrifugation (12,000$\times$ *g*, 6 min, 4 °C), the solvent was discharged, the pellet was dried with a Speed Vac for 4 min at RT and RNA was dissolved in 20 µL of RNAse-free water (Qiagen). Total RNA concentration and purity were measured with a Nanodrop$^{®}$ ND-1000 spectrophotometer. RNA integrity was confirmed by gel electrophoresis on a 1.5% agarose gel (TAE buffer) after denaturation at 70 °C for 2 min. Integrity (RIN > 7) was

additionally checked for 10% of all samples on a RNA 6000 Nano chip with an Agilent 2100 Bioanalyzer.

Subsequently, DNAse I digestion of 500 ng of undiluted RNA sample was carried out after addition of 1 μL DNAse and 1 μL 10× DNAse buffer in a volume of 10 μL. After mixing, samples were incubated for 15 min at room temperature. The reaction was stopped after addition of 1 μL 25 mM EDTA solution and inactivation of the DNase at 65 °C for 10 min.

For transcription, 500 ng μL$^{-1}$ RNA (confirmed by RiboGreen RNA quantification) were transcribed with Affinity Script MMLV transcriptase and 150 ng (in 1.5 μL H$_2$O) oligo-dT primer (5′-CCTGAATTCTAGAGCTCA(T)$_{17}$-3′). First, primer was added to 12.5 μL RNA and incubated at 65 °C for 5 min, 40 °C for 3 min, 35 °C for 3 min, 30 °C for 3 min and 25 °C for 3 min. Then, 2 μL of Affinity Script buffer, 2 μL of DTT, 1 μL dNTPs (10 mM of each) and 1 μL of Affinity Script reverse transcriptase were added and transcribed [42 °C for 60 min, 70 °C for 15 min]. In 10% of the samples, the enzyme was replaced by pure H$_2$O to monitor DNA contamination.

Specific primers for RT-QPCR analysis were designed with the sequence information available (GeneBank, NCBI) and confirmed by direct sequencing (Table 1). RT-QPCR was performed with a Mx3005 Pro (Agilent/Stratagene) using hot start polymerase (Platinum hot start Taq polymerase, Invitrogen) and SYBR Green in a 25 μL reaction volume [2 μL diluted cDNA (500 ng μL$^{-1}$), 1× PCR buffer, 3 mM MgCl$_2$, 0.2 mM dNTPs (Qiagen), 0.3 fold SYBR-Green I (Invitrogen), 0.4 μM each primer, 2 U Platinum Taq]. The cycling program involved a denaturation at 96 °C for 5 min, followed by 40 amplification cycles with a denaturation at 96 °C for 20 s, primer annealing for 20 s (Ta for each primer in Table 1) and elongation at 72 °C for 20 s. Standard, calibrator and samples were analyzed in duplicate. PCR primer efficiencies were calculated using a dilution series of a pooled sample cDNA (100 ng/μL), which was subsequently used as calibrator. Relative expression was determined by the comparative CT method (ΔΔCT) corrected for the assay efficiencies. Target expression was normalized to rpl8 as a housekeeping gene as described by Pfaffl (2001). Specificity of amplification was monitored by melting curve analysis.

**Table 1.** Specifications of QPCR assays for the relative quantification of *rpl8* (* housekeeping gene), *tcr*, *tgf β*, *tnf α*, *mhc II α* and *il-1β* providing amplicon length [bp], annealing temperature (Ta), sequence of primers, PCR efficiency (Eff) and NCBI accession number of the respective gene.

| Gene | Primer | 5′–3′ Sequence | Ta [°C] | Length [bp] | Eff. [%] | GeneBank |
|------|--------|----------------|---------|-------------|----------|----------|
| *rpl8* * | F | CTCCGCCACATTGACTTC | 64 | 197 | 94 | DQ848874 |
| | R | GCCTTCTTGCCACAGTAG | | | | |
| *Tcr* | F | ACGCCAATCACACGGTCA | 63 | 129 | 116 | AY303762 |
| | R | ATCCGAACTGCTCTCGTGG | | | | |
| *tgf β* | F | GCTACCATGCCAACTACTGC | 64 | 101 | 109 | AJ276709 |
| | R | TCCCGGGTTGTGATGCTT | | | | |
| *tnf α* | F | ATCCCCACTCCACGCTGA | 65 | 224 | 95 | FJ654645 |
| | R | CGTCCTTGCTGTCATCGTC | | | | |
| *mhc II α* | F | GATCCTCCTTCCAGTCCGAT | 63 | 140 | 105 | DQ094170 |
| | R | AATGTTGAGACTCGCTCCC | | | | |
| *il-1β* | F | CAGAAATCGCACCATGTCG | 62 | 191 | 98 | AJ295836 |
| | R | GACAACCGCAAAGTTAACCTG | | | | |

## 2.6. Statistical Analysis

Variations in sample size are a consequence of mortalities or statistical exclusion of outliers detected by standardized residuals analysis. Relative gene expression was log-transformed and analyzed for normality using the Shapiro–Wilk test and Bartlett's test for homoscedasticity. Data were subsequently analyzed with either a Kruskal–Wallis test (non-parametric) or ANOVA (parametric). If significant differences were observed, data were compared by non-parametric Dunn's or parametric Tukey's HSD. Uninfected (Pfc, Pnc, Cc) and infected groups (Pf, Pn, C) were compared using a non-parametric Wilcoxon

test or a parametric Welch two sample t-test. Differences were considered significant at $p < 0.05$. The analysis was carried out with the R software package [68].

## 3. Results

### 3.1. Mortality

Determination of LD50 after 7 d revealed a pathogen concentration of $2.65 \times 10^7$ CFU mL$^{-1}$, which was subsequently used for the experimental infection. Higher survival than expected was observed in the experimental challenge, irrespective of the group. In the Pn and Pf treatment, 0% and 8% accumulated mortality were observed after 14 d. An increased mortality of 20% was recorded for the treatment control group (C). No mortalities were observed after 1 dpi in any experimental group. In the challenge controls (no infections, Cc, Pfc, Pnc), no mortalities were observed throughout the 14-day experimental period.

### 3.2. Immune Marker Gene Expression

Even expression among samples confirmed rpl8 as a housekeeping gene for the experiment. No significant difference in gene expression was detected between uninfected fish (Pfc, Pnc, Cc), irrespective date or treatment. Gene expression of the selected immune markers revealed a similar pattern.

After infection, on 0 dpi, *tcr* expression was comparable ($p > 0.05$) between treatments (Figure 2). At 1 dpi, *tcr* expression was significantly increased in the control compared with the Pn group but not compared to the Pf treatment. At 5 dpi, expression returned to baseline levels in all treatments. No significant differences between groups were observed (one-way ANOVA, $p = 0.5910$). In the probiotic treatments, *tcr* mRNA was unchanged. In contrast, in the infected control, a significantly higher mRNA expression was observed at 1 dpi compared with 0 dpi and 5 dpi (Dunn's test, $p = 0.0136$ and $0.0096$, respectively).

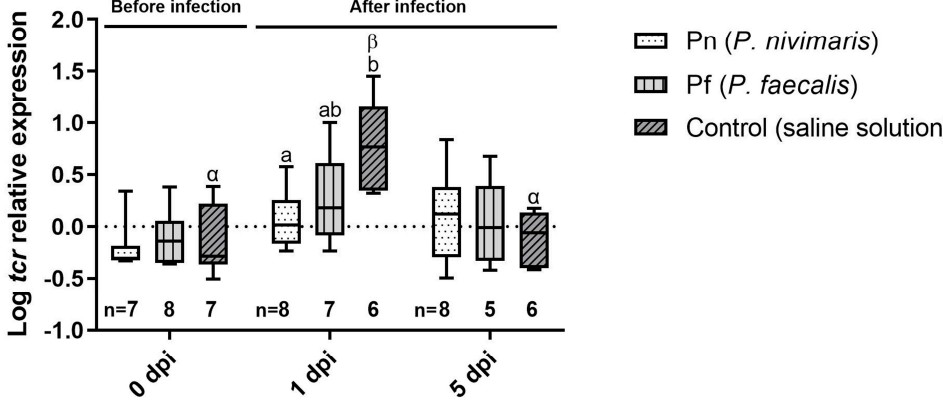

**Figure 2.** Expression of T-cell receptor (*tcr*) mRNA in the spleen of juvenile turbot at 0, 1 and 5 d post infection with *T. martimum* and previous rectal cannulation with *Psychrobacter nivimaris* (Pn), *P. faecalis* (Pf) or a NaCl saline solution control. The box indicates the median and error bars represent 5th and 95th percentiles. Significant differences between time points within a treatment are indicated by Greek letters; significant differences between treatments at a time point are indicated by Latin letters. Number of samples for each data point is given below the respective bar.

At 0 dpi, the expression of *mhc2α* was comparable between groups (Figure 3). Again, at 1 dpi, the treatment control revealed the highest gene expression compared with Pn (Tukey, $p = 0.0084$) and Pf (Tukey, $p = 0.0330$). At 5 dpi, expression in all groups returned to basal values.

At 0 dpi, mRNA expression of *il-1ß* was comparable between treatments (Figure 4). At 1 dpi, again, *il-1β* expression was higher in all infected fish compared with uninfected control fish (Wilcoxon rank sum test, $p = 0.0029$). Again, *il-1ß* was up-regulated in the infected control group compared to Pn (Dunn's test, $p = 0.0082$) and Pf (Dunn's test,

$p = 0.0162$) at 1 dpi. The expression in all treatment groups returned to baseline levels at 5 dpi (one-way ANOVA, $p = 0.2990$).

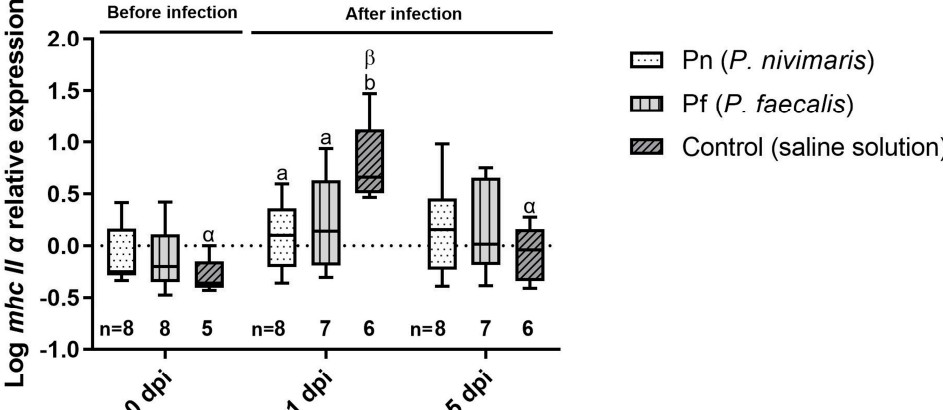

**Figure 3.** Expression of the major histocompatibility complex II alpha (*mhc 2α*) mRNA in the spleen of juvenile turbot at 0, 1 and 5 d post infection with *T. martimum* and previous rectal cannulation with *Psychrobacter nivimaris* (Pn), *P. faecalis* (Pf) or a NaCl saline solution control. The box indicates the median and error bars represent 5th and 95th percentiles. Significant differences between time points within a treatment are indicated by Greek letters; significant differences between treatments at a time point are indicated by Latin letters. Number of samples for each data point is given below the respective bar.

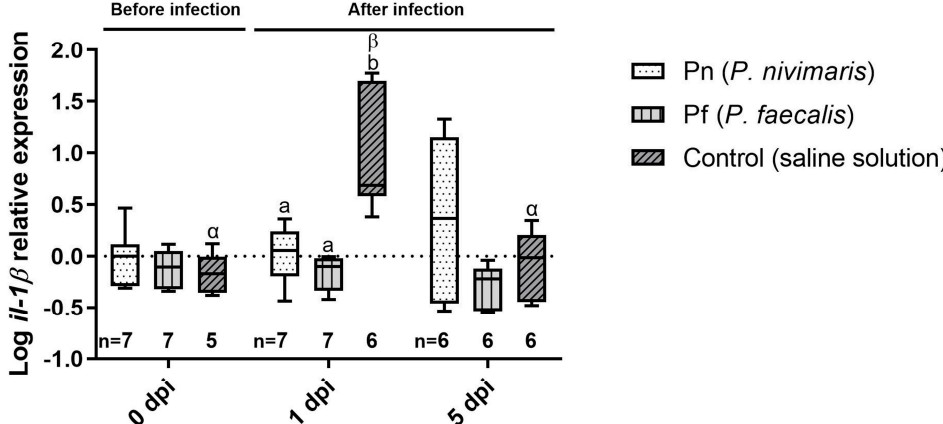

**Figure 4.** Expression of interleukin 1 beta (*il-1β*) mRNA in the spleen of juvenile turbot at 0, 1 and 5 d post-infection with *T. martimum* and previous rectal cannulation with *Psychrobacter nivimaris* (Pn), *P. faecalis* (Pf) or a NaCl saline solution control. The box indicates the median and error bars represent 5th and 95th percentiles. Significant differences between time points within a treatment are indicated by Greek letters; significant differences between treatments at a time point are indicated by Latin letters. Number of samples for each data point is given below the respective bar.

In contrast to the other target genes, *tgf* was significantly up-regulated at 0 dpi in Pn and Pf ($p = 0.0292$ and $0.0468$, respectively), indicating a slight immune stimulation in the probiotic groups (Figure 5). At 1 dpi, control fish exhibited the same increased expression as observed for the other markers over time (one-way ANOVA, $p = 0.0714$). At 5 dpi, expression levels returned to basal levels revealing no differences between treatments (one-way ANOVA, $p = 0.7830$).

The expression of *tnf α* (Figure 6) was increased only slightly at 1 dpi (one-way ANOVA, $p = 0.0520$). Only Pn revealed a lower mRNA expression than the control (Tukey multiple comparison test, $p = 0.0425$). Comparing non-infected and infected fish at 1 dpi, expression was significantly higher in all infected fish (Figure 7). Again, expression decreased to baseline levels at 5 dpi and all infected and uninfected fish revealed comparable expression.

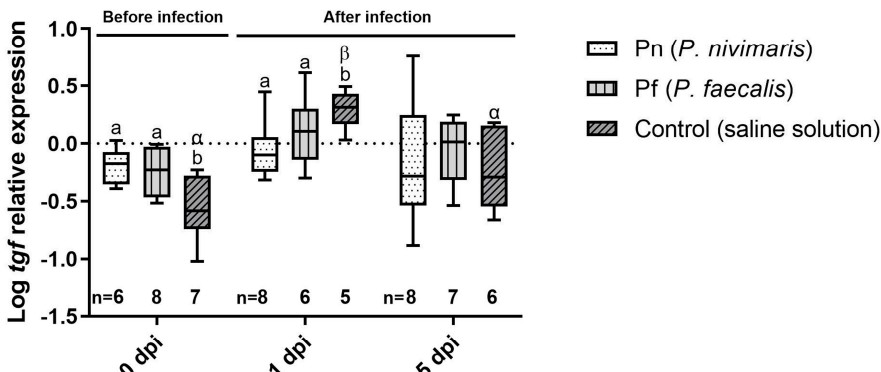

**Figure 5.** Expression of the tumor growth factor (*tgf*) mRNA in the spleen of juvenile turbot at 0, 1 and 5 d post infection with *T. martimum* and previous rectal cannulation with to *Psychrobacter nivimaris* (Pn), *P. faecalis* (Pf) or a NaCl saline solution control. The box indicates the median and error bars represent 5th and 95th percentiles. Significant differences between time points within a treatment are indicated by Greek letters; significant differences between treatments at a time point are indicated by Latin letters. Number of samples for each data point is given below the respective bar.

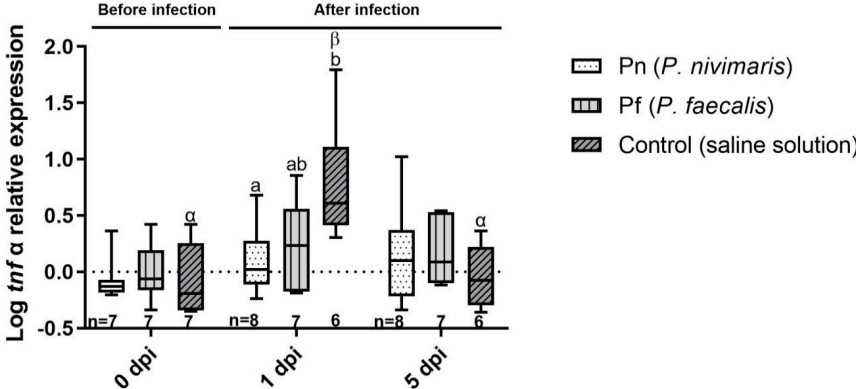

**Figure 6.** Expression of the tumor necrosis factor α (*tnfα*) mRNA in the spleen of juvenile turbot at 0, 1 and 5 d post infection with *T. martimum* and previous rectal cannulation with *Psychrobacter nivimaris* (Pn), *P. faecalis* (Pf) or a NaCl saline solution control. The box indicates the median and error bars represent 5th and 95th percentiles. Significant differences between time points within a treatment are indicated by Greek letters; significant differences between treatments at a time point are indicated by Latin letters. Number of samples for each data point is given below the respective bar.

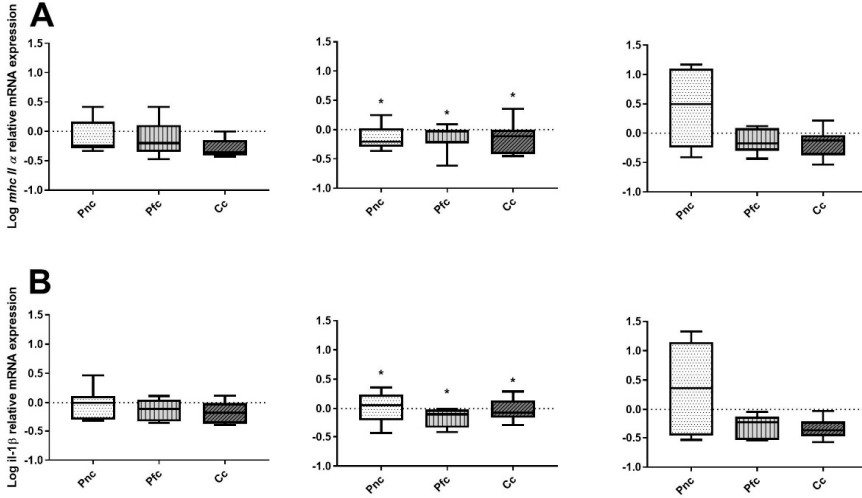

**Figure 7.** *Cont.*

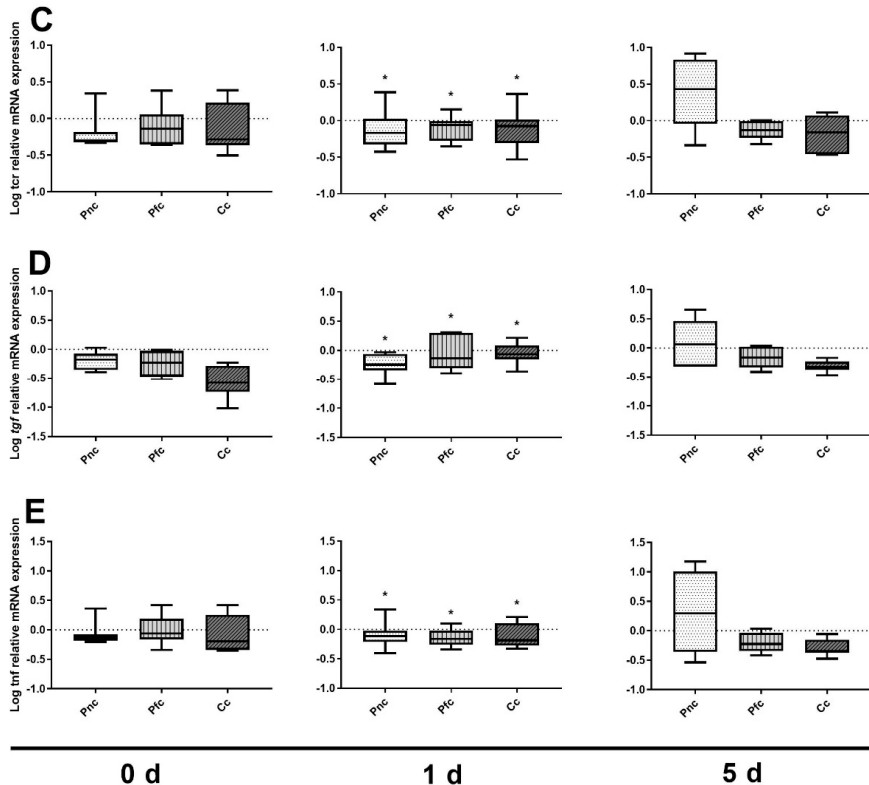

**Figure 7.** Expression of (**A**) the major histocompatibility complex II alpha (*mhc 2α*), (**B**) the interleukin 1 beta (*il-1β*), (**C**) the T-cell receptor (*tcr*), (**D**) the tumor growth factor (*tgf*) and (**E**) the tumor necrosis factor α (*tnfα*) mNA in the spleen of juvenile turbot that were not infected, serving as a treatment control at 0 d, 1 d and 5 d after rectal cannulation with *Psychrobacter nivimaris* (Pnc), *P. faecalis* (Pfc) and a NaCl saline solution control (Cc). Cannulation in these controls did not result in a changed gene expression over time. Significant differences to fish that were infected (Figures 2–6) are indicated with an asterisk. The box indicates the median and error bars represent 5th and 95th percentiles. Number of samples for each data point is given below the respective bar.

## 4. Discussion

Probiotics have been subject to intensive research in the past decade and generally show a great potential for disease management [42]. Most commercial probiotics used by the aquaculture industry are not derived from the farmed host itself, but from terrestrial sources [42]. Some evidence suggests that native probiotics are better adapted to the fish gut and do, therefore, reveal an improved colonization [69]. Furthermore, microorganisms appear to have the highest physiological performance in their natural environment [70]. Taken as a whole, it is suggested that gut microbiota coevolve with the respective host and consequently exhibit some degree of species-specific colonization of the gut. As a consequence, autochthonous candidates have been isolated and tested extensively in recent years.

In the present study, we investigated the effects of two autochthonous probiotic candidates recently isolated from wild turbot [6,43]. Both isolates showed an antagonistic effect against *T. maritimum* in a well diffusion agar assay. Furthermore, *Psychrobacter* genus was among the dominant taxa cultured [43]. Similarly, *Psychrobacter* is one of the dominant taxa in fast-growing grouper *Epinephelus coioides* [71]. Indeed, *Psychrobacter* is an integral part of the marine environment [57,72,73] including the GIT of teleosts [44,74–77]. Interestingly, *Psychrobacter* also dominates the skin mucus in Atlantic salmon *Salmo salar* [78], which may suggest a role in the first line of defense against bacterial pathogens.

For the present study, we selected two probiotic candidates and genus was identified by 16S rRNA sequencing [43]. To further improve the identification, we sequenced an

1145 bp amplicon of the gyrase b gene (gyrb) according to Zeng [63,64], which allowed identification as *Psychrobacter faecalis* (97% similarity) and *Psychrobacter nivimaris* (96% similarity). Currently, we are carrying out a whole genome sequencing project of the *Psychrobacter* candidates [43] to confirm identification at the genome level. Recently, a *Psychrobacter* sp. candidate has been isolated from Atlantic cod *Gadus morhua* [45]. Similar to our study, antagonistic effects towards *Vibrio anguillarum* and *Aeromonas salmonicida* were reported and the isolate was considered safe after in vivo testing. Recently, both candidates Pn and Pf were evaluated in a classical feeding experiment with top coated feed, revealing no adverse effects on growth performance, feed utilization and gross composition [6].

For the probiotic testing, we applied an innovative method of bacterial inoculation using a rectal cannulation technique. This technique has recently been used to by-pass deterioration of antibodies in the stomach during passive immunization [62]. It is widely believed that the stomach in gastric animals serves as a barrier to bacteria [79]. Indeed, the ubiquitous distribution of gastric acid among fish, amphibians, reptiles, birds, and mammals implies that it is evolutionarily advantageous and one of the functions hypothesized is that it inhibits pathogenic bacteria from reaching the intestine [80,81]. By-passing the stomach using the cannulation technique represents an efficient and fast method to test probiotic candidates. After probiotic administration and experimental infection, reduced mortality rates were observed in the Pn and Pf treatment groups. In a future project, we intend to establish oral delivery for the probiotic candidates via the feed.

In the present study, the two native probiotics improved survival rate after experimental infection. More importantly, administration did not increase any of the immune markers assessed at 0 dpi, except for the anti-inflammatory marker tgf, suggesting that improved survival rate was not a result of immune stimulation. Recently, it has been reported that *T. maritimum* is detectable in internal organs such as heart, kidney and brain of Atlantic salmon, suggesting that *T. maritimum* becomes systemic [82]. Therefore, it seems plausible that *Psychrobacter* may prevent internalization of *T. maritimum* by competive exclusion, congruent with the in vitro antagonism reported for both strains. Moreover, in the course of an infection with an opportunistic bacterium such as *T. maritimum* rise of secondary infections (particularly those infecting internal organs, e.g., Vibrio, Aeromonas) may result in the failure/overload of the immune system, which ultimately results in the death of the fish. In such a scenario, *Psychrobacter* may contribute in helping control ubiquitous pathogens in the GIT and thereby support its immune functioning.

After the infection, at 1 dpi, immune markers *mhc II α*, *il-1β*, *tcr*, *tgf β* and *tnf α* were up-regulated in the control but not in the probiotic-treated groups. This suggests that the probiotics acted without inducing an immune response. At the same time, due to the milder progression of the infection, probiotic treatment lead to a reduced, milder immune reaction at 1 dpi, as observed. Consistent with this mild progression of the infection, mRNA expression in all groups returned to baseline level at 5 dpi. Moreover, no mortalities were observed after 1 dpi. In grouper *Epinephelus coioides*, supplementation with *Psychrobacter* resulted in a decreased il-1ß expression, which was not observed here.

The pathogen strain used in the present study successfully infects turbot with tenacibaculosis [65,66,83]. In our study, despite a determination of LD50, relatively low mortality rates were observed. In contrast, Avendaño-Herrera et al. [65] obtained complete mortality in juvenile turbot between 1 and 10 dpi at comparable pathogen concentrations ($10^6$ to $10^8$ CFU mL$^{-1}$). However, the fish were smaller, weighing between 4 to 6 g, which may explain the deviation in results, considering a higher sensitivity of smaller fish. Indeed, Millan et al. reported 36% mortality in 20 g fish after an experimental infection with *A. salmonicida*. Similar to our study, an LD50 had been determined previously [84].

After experimental infection, at 0 dpi, *tgf* was increased in both probiotic groups (Pfc, Pnc). This cytokine is activated by regulatory T-cells. It consequently inhibits the action of other T-cells, thereby acting as an immunosuppressor [85]. Its up-regulation may, therefore, explain, at least in part, the immunosuppression observed in the probiotic groups. In turbot, Millan et al. [84] reported an increase of *il-1ß* and *tnf α* in the spleen as

well as an up-regulation of *mhc II α* in the liver of *Aeromonas salmonicida* infected turbot. Similarly, formalin-inactivated *Edwardsiella tarda* triggers an up-regulation of *mhc II*, *il-1ß*, *tcr*, and *tnf α* [86]. Sun et al. [52] reported increased phagocytic index, phagocytic activity, superoxide dismutase activity and serum complement component 3 (*c3*) and 4 (*c4*) after administration, suggesting immune stimulatory effects after treatment with *Psychrobacter* sp. However, such effects were not observed here.

**Author Contributions:** Conceptualization, S.W. and B.C.; methodology, S.W. and B.C.; validation, S.W., B.C. and F.B.; formal analysis, E.K. and F.B.; investigation, F.B., K.M.W., M.M. and R.A.; data curation, F.B.; writing—original draft preparation, S.W. and F.B.; writing—review and editing, B.C. and R.A.; visualization, F.B.; supervision, S.W. and B.C.; project administration, S.W. and B.C.; funding acquisition, S.W. and B.C. All authors have read and agreed to the published version of the manuscript.

**Funding:** This work was partially funded by FCT/DAAD Acções Integradas Luso-Alemãs (references 441.00 DAAD and 57051461). R.A., M.M. and B.C. were supported by FCT (SFRH/BD/89457/2012, SFRH/BD/108243/2015 and IF/00197/2015). F.B. was supported by the ERASMUS+ Traineeship Programme of the European Union.

**Institutional Review Board Statement:** The study was carried out in compliance with the the EU Directive 2010/63/EU and Portuguese legislation and approved by the CIIMAR Animal Welfare Committee (0421/000/000/2020).

**Informed Consent Statement:** Not applicable.

**Data Availability Statement:** Data will be made available upon request.

**Acknowledgments:** The *T. maritimum* strain ACC13.1 was kindly provided by Alicia E. Toranzo (Departamento de Microbiologia y Parasitologia, Facultad de Biologıa, University of Santiago de Compostela, Spain)—many thanks for this support.

**Conflicts of Interest:** The authors declare no conflict of interest.

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
