# Peer review of "Two Probiotic Candidates of the Genus Psychrobacter Modulate the Immune Response and Disease Resistance after Experimental Infection in Turbot (Scophthalmus maximus, Linnaeus 1758)"

_fishes, doi:10.3390/fishes8030144_

Round 1

Reviewer 1 Report

The topic of this article is exploring that two probiotic candidates of the genus Psychrobacter modulate the immune response and disease resistance after experimental infection in turbot. This paper mainly describes the survival and the mRNA expression of immune marker genes of turbot which pretreated with two autochthonous candidates P. nivimaris and P.Faecalis following by treating with T.maritimum. While the subject is interesting, I am sorry to decide to reject. The main reasons include:

The results of this article are all the mRNA expression of immune genes in the spleen of turbot, which cannot fully prove the conclusion. And the data in this article is insufficient.

There is a problem with the significance analysis in the figure 7. The data to be compared must appear in the same figure.

Author Response

We greatly acknowledge the time and effort of the reviewer as well as the editor. This greatly helped us to revise and improve the manuscript. We revised the manuscript as suggested. All changes in tracked changes mode can be found in the attached file of the manuscript. Our response and the line numbers below refer to this tracked changes version of our manuscript.

Reviewer: 1
The topic of this article is exploring that two probiotic candidates of the genus Psychrobacter modulate the immune response and disease resistance after experimental infection in turbot. This paper mainly describes the survival and the mRNA expression of immune marker genes of turbot which pretreated with two autochthonous candidates P. nivimaris and P.Faecalis following by treating with T.maritimum. While the subject is interesting, I am sorry to decide to reject. The main reasons include:

The results of this article are all the mRNA expression of immune genes in the spleen of turbot, which cannot fully prove the conclusion. And the data in this article is insufficient.

We do not agree with this statement. We carefully describe the results, namely a reduced mortality and a difference in the expression of 5 key marker genes of the immune response, comparing fish that received probiotic administration and control fish (all five markers are increased in the control due to the infection whereas markers do not increase in the probiotic group). Considering that the probiotic groups reveal a reduced mortality we suggest that this is based on comparative exclusion as no immune stimulation (of the 5 markers) was observed. We reformulated this conclusion in the discussion:

Overall, our results indicate that Pn and Pf act by competitive exclusion rather than immune stimulation

As a general conclusion, we believe the two candidates have a potential in disease prophylaxes (L34-35). Undoubtedly further research is required before putting into practice (L417-418: In a future project, we intend to establish oral delivery for the probiotic candidates via the feed and confirm the observations reported here). We formulated this conclusion more carefully: This highlights a potential application of P. nivimaris and P. faecalis in disease prophylaxis, but further research is required.

We clearly describe the experiment, present the results. As stated, we reformulated our conclusions more carefully.

There is a problem with the significance analysis in the figure 7. The data to be compared must appear in the same figure.

The Fig 7 presents the non-infected control supplementing the outcome of the experiment. It demonstates that the is no change in expression in fish that were not infected. Differences (pairwise comparison) to the respective fish that were infected are marked with an asterisk. This is clearly described in the results. This figure shows that expression of non-infected fish did not change over time.

Reviewer 2 Report

1.      Turbot treated with control treatment resulted in 20% mortality post infection with Tenacibaculum maritimum. It looks like this pathogen is not a lethal bacterium for turbot. The authors need to find a more lethal pathogen to treat turbot.

2.      Line 163 ppm is a wrong term to express the salinity.

3.      The authors need to describe how to challenge bacteria.

4.      Line 298 After infection, at 0 dpi, tcr expression was comparable between treatments (Fig. 2). What does “comparable” mean?

5.      Authors should explain what are a, b, --- or α,βin the figures.

Author Response

We greatly acknowledge the time and effort of the reviewer as well as the editor. This greatly helped us to revise and improve the manuscript. We revised the manuscript as suggested. All changes in tracked changes mode can be found in the attached file of the manuscript. Our response and the line numbers below refer to this tracked changes version of our manuscript.

Reviewer: 1
.      Turbot treated with control treatment resulted in 20% mortality post infection with Tenacibaculum maritimum. It looks like this pathogen is not a lethal bacterium for turbot. The authors need to find a more lethal pathogen to treat turbot.

Tenacibaculum is a mayor problem in turbot aquaculture [24-28]. Therefore we used it in our study. We also discuss the relatively low mortality observed (L434-444). Also, we performed an LD50 test for the selection of the actual concentration of the pathogen in our study. Still, the reviewer is right, 20% is very low, we targeted 50% LD50)

  1. Line 163 ppm is a wrong term to express the salinity.

As described, we used a handprobe to determine salinity. Unfortunately, it did not determine PSU.

  1. The authors need to describe how to challenge bacteria.

The challenge is described in detail (L208-215). As described, some details on the preparation of the bath are already described in the chapter before, e.g. isolation of bacteria (L185-207)

  1. Line 298 After infection, at 0 dpi, tcr expression was comparable between treatments (Fig. 2). What does “comparable” mean?

Comparable means no significant difference. This was specified: was comparable (p > 0.05) between treatments

  1. Authors should explain what are a, b, --- or α,βin the figures.

This is described in detail in the legends: Significant differences between time points, within a treatment, are indicated by Greek letters, significant differences between treatments at a time point are indicated by Latin letters. Number of samples for each data point is given below the respective bar.

Reviewer 3 Report

The work presented is interesting and the authors demonstrated the working mechanism of the two potential probiotic bacteria. However, there are issues related to the work:

The manuscript has not been properly proofread before submission. Throughout the whole document, many mistakes can be seen.

-          Superscripts and subscripts of units

-          Italicize the species names of bacteria and fish in the discussion.

Please check and update accordingly.

Is T. maritimum common in RAS? Can it be prevented using UV or ozone?

The authors cited references to suggest flatfish is susceptible to T. maritimum infection (line 64-65). Are there any outbreak reports? Is the infection related to temperature? Estimated loss and the mortality rates? If yes, please provide more information to help the future readers to understand the situation.

Line 156 – 157: “” Do the authors intend to show the configuration of the RAS? Or is it a typo? Please provide more information about the RAS used in the study.

According to Fig 1, the 1st sampling was conducted 5d after the 2nd cannulation. However, line 182 suggested the sampling was conducted 4d post cannulation. Please clarify.

What is the rationale for having two cannulation? I think more information provided would be beneficial to the future readers.

Were the fish fed before/after cannulation/infection? There was no relevant description on this issue. 

For the interest of future readers, I think it would be nice to show the devices/progress of cannulation, as it could be a more accurate mean to test the effect of probiotic bacteria.

Treatment groups and treatment control were used in line 156-157. The description provided there gave me an idea that three experimental groups were used. What is the meaning of challenge controls of non-infected fish in line 211? Is it an extra treatment group? Again, what is “challenge controls” in line 213? Please clarify/ rename/standardize the names of the treatment groups.

Did the authors attempt to recover Psychrobacter spp. from the treated fish? Some of the probiotic bacteria cannot colonize fish gut and they are just “passengers”. If possible, I think it will be interesting to attempt that. Right now, according to the the presented data, the readers cannot tell how long they can stay.

According to the references provided by the authors, it seemed that T. maritimum causes infections on fish surface e.g. gill, mouth, fin and tail. It is interesting that the hindgut administered probiotic can inhibit T. maritimum. I think the authors should provide more discussion on this observation.

I understand that the T. maritimum was donated by Prof. Toranzo. But more information related to the isolate can be provided in the manuscript e.g. How lethal is the T. maritimum isolate? Where was is isolated? From skin or intestine?

Although there was no statistical analysis, the reduction in mortality of T. maritimum challenge was rather significant. A possible explanation of the lowered mortality could be the no. of passage (subculture). The authors did not mention the temperature of the tanks during LD50 determination. The optimal temperature of turbot is 14 - 18 ºC, and the optimal temp of T. maritimum is 30ºC. Would that be the reason for the lower mortality observed? As the RAS was arrested during the LD50 determination, would the water temperature be affected and lower the activities of the bacteria? I believe more information would be needed to explain the observation.

Probiotic bacteria might be able to improve FCR/growth. Do you think those probiotics used in the present study can help turbot too?

Author Response

We greatly acknowledge the time and effort of the reviewer as well as the editor. This greatly helped us to revise and improve the manuscript. We revised the manuscript as suggested. All changes in tracked changes mode can be found in the attached file of the manuscript. Our response and the line numbers below refer to this tracked changes version of our manuscript.

Reviewer: 3
The work presented is interesting and the authors demonstrated the working mechanism of the two potential probiotic bacteria. However, there are issues related to the work:

The manuscript has not been properly proofread before submission. Throughout the whole document, many mistakes can be seen.

-          Superscripts and subscripts of units

-          Italicize the species names of bacteria and fish in the discussion.

Please check and update accordingly.

Done, please refer to the corrected version.

Is T. maritimum common in RAS? Can it be prevented using UV or ozone?

To our knowledge, there is no study on this subject. Still, one would expect from other studies that ozone and dependent on the setup also UV reduces the risk of infection. 

The authors cited references to suggest flatfish is susceptible to T. maritimum infection (line 64-65). Are there any outbreak reports? Is the infection related to temperature? Estimated loss and the mortality rates? If yes, please provide more information to help the future readers to understand the situation.

 Outbreaks are a particular problem in Spain but the infection involves a variety of species including Solea senegalensis, Solea solea and even salmonids. Due to the high temperature range (15-34°C) it occurs particularly in summer on the Iberian Peninsula. Obviously, elevation of temperature stresses cultured fish which turns them susceptible towards diseases. This information was included in the introduction:

It is particularly important in turbot farming in Spain and Portugal and numerous outbreaks have been reported [30, 31]. T. maritimum is considered an opportunistic pathogen where the outbreak of disease is frequently associated with increasing temperatures, salinity and reduced water quality [31].

Line 156 – 157: “” Do the authors intend to show the configuration of the RAS? Or is it a typo? Please provide more information about the RAS used in the study.

We added a short description of the RAS:

Each RAS was composed of tanks and a sump and was run with a water turnover of 2 volumes/h and 2-4% water exchange. The sump included a biological and mechanical filter unit. The water from the tanks was first directed to a mechanical filter, which comprised a series of trays with cloths retaining  the debris. Then, the water was directed into a reservoir with the biological filter comprising bio carriers in constant movement and strong aeration. The filter water is then finally pumped into the tanks. 

According to Fig 1, the 1st sampling was conducted 5d after the 2nd cannulation. However, line 182 suggested the sampling was conducted 4d post cannulation. Please clarify.

 Corrected: First sampling (0 dpi) was carried out five days after the second rectal cannulation.

What is the rationale for having two cannulation? I think more information provided would be beneficial to the future readers.

This has been described in the abstract (After rectal administration to maximise colonization by-passing digestion in the stomach) and the introduction (To standardize the administration, avoid deterioration during the passage of the gastrointestinal tract (GIT) and support the colonization in the GIT, we administrated the probiotics via rectal cannulation as previously described [62] and is later on discussed in further detail:

For the probiotic testing, we applied an innovative method of bacterial inoculation using a rectal cannulation technique. This technique has recently been used to by-pass deterioration of antibodies in the stomach during passive immunization [62]. It is widely believed that the stomach in gastric animals serves as a barrier to bacteria [79]. Indeed, the ubiquitous distribution of gastric acid among fish, amphibians, reptiles, birds, and mammals implies that it is evolutionarily advantageous and one of the functions hypothesized is that it inhibits pathogenic bacteria from reaching the intestine [80, 81]. By-passing the stomach using the cannulation technique represents an efficient and fast method to test probiotic candidates.

Were the fish fed before/after cannulation/infection? There was no relevant description on this issue. 

This information has been included:

Feeding was suspended 24 h before the cannulation and resumed approximately 15 h after the treatment.

For the interest of future readers, I think it would be nice to show the devices/progress of cannulation, as it could be a more accurate mean to test the effect of probiotic bacteria.

The details on the cannulation are described in the text.

Treatment groups and treatment control were used in line 156-157. The description provided there gave me an idea that three experimental groups were used. What is the meaning of challenge controls of non-infected fish in line 211? Is it an extra treatment group? Again, what is “challenge controls” in line 213? Please clarify/ rename/standardize the names of the treatment groups.

To monitor the effect of the cannulation alone we assessed non-infected fish from all three cannulation groups. These controls (control to the bacterial challenge) are termed (Pfc, Pnc, Cc).  This has been described in the M&M (L221): In addition, challenge controls of non-infected fish (Pfc, Pnc, Cc), that received the cannulation, were reared in a separate RAS. As described in the text and in fig 7 there were no mortalities observed and gene expression was not affect by the cannulation.

Did the authors attempt to recover Psychrobacter spp. from the treated fish? Some of the probiotic bacteria cannot colonize fish gut and they are just “passengers”. If possible, I think it will be interesting to attempt that. Right now, according to the the presented data, the readers cannot tell how long they can stay.

Unfortunately, we did not study this. Still, we want to consider this aspect in a future experiment. 

According to the references provided by the authors, it seemed that T. maritimum causes infections on fish surface e.g. gill, mouth, fin and tail. It is interesting that the hindgut administered probiotic can inhibit T. maritimum. I think the authors should provide more discussion on this observation.

Indeed, there is lot of debate on the actual cause of dead in Tenacibaculosis. Internalization of the bacterium has been reported. Therefore, Psychrobacter may help preventing that T. maritimum becomes systemic. Since we did not observe an indication of an immune stimulation we believe this is the most probable scenario. Additionally, in the course of an infection with an opportunistic bacterium such as T. maritimum rise of secondary infections (particularly those infecting internal organs, e.g. Vibrio, Aeromonas) may result in the failure/overload of the immune system and ultimately result in the death of the fish. In such a scenario, Psychrobacter may contribute in helping control ubiquitous pathogens in the GIT and thereby support its immune functioning. We believe that a continuous inoculation of the epidermal mucus is less probable since studies have shown that the microbiome of the epidermis is quite different from the microbiome of the intestinal tract, suggesting that inoculation is is not the best hypothesis. We changed the discussion with regard to these hypothesis:

In the present study, the two native probiotics improved survival rate after experimental infection. More importantly, administration did not increase any of the immune markers assessed at 0 dpi, except for the anti-inflammatory marker tgf, suggesting that improved survival rate was not a result of immune stimulation. Recently, it has been reported that T. maritimum is detectable in internal organs such as heart, kidney and brain of Atlantic salmon, suggesting that T. maritimum becomes systemic [82]. Therefore, it seems plausible that Psychrobacter may prevent internalization of T. maritimum by competive exclusion, congruent with the in vitro antagonism reported for both strains. Also, in the course of an infection with an opportunistic bacterium such as T. maritimum rise of secondary infections (particularly those infecting internal organs, e.g. Vibrio, Aeromonas) may result in the failure/overload of the immune system, which ultimately results in the death of the fish. In such a scenario, Psychrobacter may contribute in helping control ubiquitous pathogens in the GIT and thereby support its immune functioning.

I understand that the T. maritimum was donated by Prof. Toranzo. But more information related to the isolate can be provided in the manuscript e.g. How lethal is the T. maritimum isolate? Where was is isolated? From skin or intestine?

As described we determined the LD50 diretly before the actual experiment. Still, we only observed 20% mortality (instead of the targeted 50%). We considered the growth of the fish in 14 d as minor but this might have contributed to the reduced mortality in the actual experiment. Also, fish in the LD50 test received less acclimatization time (14 d).

  1. maritimum was isolated from skin lesion. We included the information in the manuscript.

Although there was no statistical analysis, the reduction in mortality of T. maritimum challenge was rather significant. A possible explanation of the lowered mortality could be the no. of passage (subculture). The authors did not mention the temperature of the tanks during LD50 determination. The optimal temperature of turbot is 14 - 18 ºC, and the optimal temp of T. maritimum is 30ºC. Would that be the reason for the lower mortality observed? As the RAS was arrested during the LD50 determination, would the water temperature be affected and lower the activities of the bacteria? I believe more information would be needed to explain the observation.

 The room temperature was controlled by an air condition. There was no difference in temperature between the determination of the LD50 and the actual experiment

Probiotic bacteria might be able to improve FCR/growth. Do you think those probiotics used in the present study can help turbot too?

Interesting question. We did perform a study where we replaced fishmeal with soybean meal. In one group we additionally feed the probiotics. We did not observe an improved growth or feed conversion in the probiotic group compared to the soybean meal control.

Round 2

Reviewer 1 Report

With authors' point-by-point response, I could get the picture of Fig.7. The figure legends and data description of Fig7 should be further revised to avoid misunderstanding. 

Author Response

We revised the legend accordingly:

Figure 7. Expression of A) the major histocompatibility complex II alpha (mhc 2α), B) the interleukin 1 beta (il-1β), C) the T-cell receptor (tcr), D) the tumor growth factor (tgf) and E) the tumor necrosis factor α (tnfα) mNA in the spleen of juvenile turbot that were not infected, serving as a treatment control at 0 d, 1 d and 5 d after rectal cannulation with Psychrobacter nivimaris (Pnc), P. faecalis (Pfc) and a NaCl saline solution control (Cc). Cannulation in these controls did not result in a changed gene expression over time. Significant differences to fish that were infected (Fig. 2-6) are indicated with an asterisk. The box indicates the median and error bars represent 5th and 95th percentiles. Number of samples for each data point is given below the respective bar.